# Sample Size Calculations in Simple Linear Regression: A New Approach

**DOI:** 10.3390/e25040611

**Published:** 2023-04-03

**Authors:** Tianyuan Guan, Mohammed Khorshed Alam, Marepalli Bhaskara Rao

**Affiliations:** 1College of Public Health, Kent State University, 750 Hilltop Drive, Kent, OH 44240, USA; 2Department of Environmental Health and Public Health Sciences, University of Cincinnati, 160 Panzeca Way, Cincinnati, OH 45221, USA

**Keywords:** least squares estimator, level, power, unconditional distribution

## Abstract

The problem tackled is the determination of sample size for a given level and power in the context of a simple linear regression model. The standard approach deals with planned experiments in which the predictor X is observed for a number n of times and the corresponding observations on the response variable Y are to be drawn. The statistic that is used is built on the least squares’ estimator of the slope parameter. Its conditional distribution given the data on the predictor X is utilized for sample size calculations. This is problematic. The sample size n is already presaged and the data on X is fixed. In unplanned experiments, in which both X and Y are to be sampled simultaneously, we do not have data on the predictor X yet. This conundrum has been discussed in several papers and books with no solution proposed. We overcome the problem by determining the exact unconditional distribution of the test statistic in the unplanned case. We have provided tables of critical values for given levels of significance following the exact distribution. In addition, we show that the distribution of the test statistic depends only on the effect size, which is defined precisely in the paper.

## 1. Introduction

Multiple regression is one of the core methodologies in statistics. Power computation and sample size determination have become integral part of many research proposals submitted for funding. Funding agencies such as UKRI (UK Research and Innovation) and NIH (National Institutes of Health) have been demanding sample size calculations in all prospective proposals. Regression has a long history dating back to Galton [1]. Horton and Switzer [2] reported that 51% of research articles published in the New England Journal of Medicine during May 2004 have Multiple Regression as one of the methods used. The figure for power analysis is at 39%. 

In this paper, we focus on power computation in the context of simple linear regression. The current approach in power computations lacks justification. We will point out difficulties in this setting [3]. 

Simple linear regression is ubiquitous in pediatric clinical diagnostics. The model sets standards for normal growth in children on several metrics [4]. As an illustration, a pediatrician wants to check whether the lung function of a 13-year-old patient is normal. Data is to be collected on healthy subjects in the age range 12–14 years with response,
Y = FEV (Forced Expiratory Volume) 
and predictor,
X = Height,
which is an example of an unplanned experiment.

In order to trust the model, we need to decide on the sample size, which in turn, depends on the level of significance, power, and effect size. 

First, we investigate the setting under the simple linear regression paradigm. The model has two entities X, the predictor, and Y, the response variable. It is stated as
$$Y|X\sim N\left(\beta_{0}+\beta_{1}X,{\;\sigma }^{2}\right)$$
for some $\beta_{0}$, $\beta_{1},$ and $\sigma^{2}>0$. The null hypothesis of interest is $H_{0}:\beta_{1}=0$ against the alternative $H_{1}:\beta_{1}\neq 0$. What should be the required sample size, n, for a given level of significance α, power 1-β, and at the alternative value A of $\beta_{1}$. Let (X_1_, Y_1_), (X_2_, Y_2_), …, (X_n_, Y_n_), be a potential sample for the testing problem. Let ${\hat{\beta }}_{1}$ be the least squares estimator of $\beta_{1}$, i.e.,
$${\hat{\beta }}_{1}=\frac{S_{\mathrm{XY}}}{S_{\mathrm{XX}}}$$
where
$$S_{\mathrm{XY}}=\overset{n}{\underset{i=1}{\sum }}\left(X_{i}-\bar{X}\right)\left(Y_{i}-\bar{Y}\right)$$
and
$$S_{XX}=\overset{n}{\underset{i=1}{\sum }}{\left(X_{i}-\bar{X}\right)}^{2}$$

Let RSS be the residual sum of squares, i.e.,
$$\mathrm{RSS}=\overset{n}{\underset{i=1}{\sum }}{\left[\left(Y_{i}-\bar{Y}\right)-{\hat{\beta }}_{1}\left(X_{i}-\bar{X}\right)\right]}^{2}$$

For testing the null hypothesis H0, the following test statistic is used:$$T=\left({\hat{\beta }}_{1}\sqrt{S_{\mathrm{XX}}}\right)/\left(\sqrt{\mathrm{RSS}/\left(n-2\right)}\right).$$

Under the null hypothesis, conditioned on the X-data, T has a t-distribution with n − 2 degrees of freedom. Under the alternative value $\beta_{1}$ = A, T has a non-central t-distribution with degrees of freedom n − 2, and non-centrality parameter $\lambda =\left|A\right|*\sqrt{\mathrm{Sxx}\;}/\sigma$.

We reject the null hypothesis if and only if $\left|T\right|>t_{n-2,1-\frac{\alpha }{2}}$ where $t_{n-2,1-\frac{\alpha }{2}}$ is such that the area to the left of Student’s t-curve on (n − 2) degrees of freedom is 1 − α/2. 

The power formula is given by
$$\mathrm{Power}\left(A\right)=\mathrm{Pr}\left({\mathrm{Reject}\;H}_{0}|\beta_{1}=A\right)=\mathrm{Pr}\left(\left|T\right|>t_{n-2,1-\frac{\alpha }{2}}|\beta_{1}=A\right).$$

We can set power equal to 1-β and solve for n. This would work as long as we know what $\lambda =\left|A\right|*\sqrt{S_{\mathrm{XX}}\;}/\sigma$ is. This would require knowledge of the alternative value of β_1_, $\sigma^{2}$, and $S_{\mathrm{XX}}$. We will not know what $S_{\mathrm{XX}}$ is, prior to data collection, in the unplanned experiments. Equivalently, one should spell out what λ is. This is a tall order. Adcock [5] recognized these problems. Some software and textbooks assume that $\left(1/n\right)\sum_{i=1}^{n}{\left(X_{i}-\bar{X}\right)}^{2}$ is known. For example, the software PASS [6] and nQuery [7] proceed this way. To overcome these difficulties, we proceed with deriving the exact unconditional distribution of a variant of T. This requires a knowledge of the distribution of X. Let $\sigma_{X}^{2}$ be the variances of X.

Modify the test statistic.
(1)$$T={\hat{\beta }}_{1}*{\hat{\sigma }}_{X}/\hat{\sigma },\;$$
where ${\hat{\sigma }}^{2}=\mathrm{RSS}/\left(n-2\right)$ ${\mathrm{and}\hat{\sigma }}_{X}^{2}=S_{\mathrm{XX}}/\left(n-1\right).$

We obtain the unconditional distribution of T under β_1_ = 0 as well as under β_1_ = A ≠ 0. We assume X~N (µ_x_, $\sigma_{X}^{2}$), both parameters unknown. Under this assumption, the distribution of T is derived.

In due course, we will show the distribution of T when β_1_ = A ≠ 0 depends only on δ = $\left|A\right|{*\sigma }_{X}/\sigma$, which we can deem as the effect size.

The five-parameter model now is:$$Y|X\sim N\left(\beta_{0}+\beta_{1}X,{\;\sigma }^{2}\;\right)$$
$$X\sim N({µ}_{x},\;\sigma_{X}^{2}).$$

Note that the vector (X, Y) has a bivariate normal distribution.

The paper is organized as follows. In Section 2, we provide a literature review. In Section 3, we outline the main results. We derive the unconditional distribution of T under the null hypothesis in Section 3.1. In Section 3.2, we calculate critical values using the main results. In Section 3.3, we lay out the sample sizes required for a given level, power, and effect size $\delta \;=\left|A\right|{*\sigma }_{X}/\sigma$. In Section 4, we summarize the results and draw conclusions. The computational details along with the R code [8] are presented in the Appendix A.

## 2. Literature Review 

Ryan [3] has pointed out difficulties in power calculations in the environment of simple linear regression. The problem is how we handle the predictor X. Adcock [5] has looked at some possible scenarios. One scenario is that the investigator knows the X_i_-values (deterministic) for every sample size n. In such a case, the test statistic
(2)$$\left({\hat{\beta }}_{1}\sqrt{S_{\mathrm{XX}}}\right)/\left(\sqrt{\mathrm{RSS}/\left(n-2\right)}\right)$$
is eminently usable for power calculations. Its (conditional) null and non-null distributions have been worked out explicitly. The conditional approach is also followed by Dupont et al. [9], Draper et al. [10], Hsieh et al. [11], Maxwell [12], and Thigpen [13]. 

As an alternative to the test statistic (2), we can build a test based on the sample correlation coefficient $\hat{\rho }$ [14], under the joint normality of X and Y. The null and non-null distributions of the underlying test statistic based on $\hat{\rho }$ have been worked out explicitly. In our consulting work, many researchers prefer to use the test based on ${\hat{\beta }}_{1}$. It is a choice between causality and association [3,14,15,16,17,18,19,20,21]. The hypotheses H_0_: $\beta_{1}$ = 0 and H_0_: ρ  = 0 under bivariate normality are equivalent, but the test statistics are different. It is easy to determine sample size under the correlation context [14]. However, this sample size cannot be offered for testing the hypothesis on the slope. The power is less. In other words, test hopping is not permissible; i.e., they are two different tests with distinct power functions.

## 3. Outline of Results

We will now derive the unconditional distribution of ${\hat{\beta }}_{1}$, which will be instrumental in sample size calculations. We use the test statistic T = ${\hat{\beta }}_{1}*{\hat{\sigma }}_{X}/\hat{\sigma }$.

Under the null hypothesis β_1_ = 0, we show that
$$T^{2}\sim \frac{\left(n-2\right)}{\left(n-1\right)}*\frac{W_{1}W_{4}}{W_{2}W_{3}},$$
where $W_{1}\sim \chi_{1}^{2}$, $W_{2}\sim \chi_{n-1}^{2}$, $W_{3}\sim \chi_{n-2}^{2}$ and $W_{4}\sim \chi_{n-1}^{2}$, with the W_i_ values being mutually independent. It follows implicitly that
$$T\;\sim \;\sqrt{\left(n-2\right)∕\left(n-1\right)}*U_{1}*U_{2}/U_{3}$$
with $U_{1},\;U_{2},\;U_{3}$ independently distributed, $U_{1}\sim t_{n-1},\;U_{2}\sim \chi_{n-1}$, and $U_{3}\sim \chi_{n-2}$, and where $\chi_{n-1}$ is the χ distribution with $\left(n-1\right)$ degrees of freedom

We use this result to obtain the critical values of the test based on T, for given levels. For power and sample size computations, we need the distribution of T for any given value of β_1_. The distribution depends on the alternative values of β_1_, $\sigma_{X}^{2}$ and σ^2^. It turns out that the distribution depends only on δ = $\beta_{1}{*\sigma }_{X}/\sigma$, which we can deem as the effect size. The specification of δ facilitates computation of power. Despite all these deliberations, no magic explicit formula for power surfaces. Knowing the distribution of T^2^ when δ is spelled out, the pain is eased a little bit. 

### 3.1. Distributional Results

In this section, we will derive the distribution of T of (1) unconditionally. The following series of steps will give the desired result. 

Given X_1_, X_2_, …, X_n_, ${\hat{\beta }}_{1}$ has a normal distribution with mean $\beta_{1}$ and variance $\sigma^{2}/S_{XX}$ and ${\hat{\beta }}_{1}$ and RSS are independent.Uncoditionally, $RSS/\sigma^{2}\sim \chi_{n-2}^{2}$$S_{XX}/\sigma_{X}^{2}\sim \chi_{n-1}^{2}$.RSS and $S_{XX}$ are independent.

More generally, we obtain the distribution of $T=\left({\hat{\beta }}_{1}-\beta_{1}\right){\hat{\sigma }}_{X}/\hat{\sigma }$ for a given value of β_1_. 

The joint density function of ${\hat{\beta }}_{1}$ and $S_{XX}$:$$\begin{array}{l} g\left({\hat{\beta }}_{1},S_{XX}\right)=\frac{\sqrt{S_{XX}}}{\sqrt{2\pi }\sigma }*\exp\left(-\frac{S_{XX}}{2\sigma^{2}}{\left({\hat{\beta }}_{1}-\beta_{1}\right)}^{2}\right)*\frac{1}{Г\left(\frac{n-1}{2}\right)*2^{\frac{n-1}{2}\;}}\exp\left(-\frac{S_{XX}}{2*\sigma_{X}^{2}}\right)*{\left(\frac{S_{XX}}{\sigma_{X}^{2}}\right)}^{\frac{n-1}{2}-1}*\left(\frac{1}{\sigma_{X}^{2}}\right),\\ -\infty <{\hat{\beta }}_{1}<\infty ,\;0<S_{XX}<\infty \end{array}$$

The (unconditional) marginal density of ${\hat{\beta }}_{1}$ is given by:$$\begin{array}{ll} f\left({\hat{\beta }}_{1}\right) & =\frac{1}{2^{\left(\frac{1}{2}\right)}*2^{\frac{n-1}{2}}*\sqrt{\pi }*Г\left(\frac{n-1}{2}\right)*\sigma *{\left(\sigma_{X}^{2}\right)}^{\frac{n-1}{2}}}\int_{0}^{\infty }S_{XX}{}^{\frac{n}{2}-1}*\exp\left(-\frac{S_{XX}}{2}\left[\;\frac{{\left(\hat{\beta_{1}}-\beta_{1}\right)}^{2}}{\sigma^{2}}+\frac{1}{\sigma_{X}^{2}}\right]\right)dS_{XX}\\ & =\frac{Г\left(\frac{n}{2}\right)}{\sqrt{\pi }*Г\left(\frac{n-1}{2}\right)*\sigma *{\left(\sigma_{X}^{2}\right)}^{\frac{n-1}{2}}}{\left(\frac{1}{\frac{1}{\sigma_{X}^{2}}+\frac{{\left({\hat{\beta }}_{1}-\beta_{1}\right)}^{2}}{\sigma^{2}}}\right)}^{\frac{n}{2}}\\ & =\frac{\sigma_{X}}{B\left(\frac{1}{2},\frac{n-1}{2}\right)*\sigma }*\frac{1}{{\left(1+\frac{{\left({\hat{\beta }}_{1}-\beta_{1}\right)}^{2}*\sigma_{X}^{2}}{\sigma^{2}}\right)}^{\frac{n}{2}}},\;\;\;\;\;\;\;\;\;\;\;\;\;-\infty <{\hat{\beta }}_{1}<\infty \end{array}$$

Some properties of this density are clear to observe. For example, the distribution is symmetric around the true value $\beta_{1}$. If n = 2, the distribution is Cauchy. In addition,
$$\left(\frac{\sigma_{X}}{\sigma }\right)*\left({\hat{\beta }}_{1}-\beta_{1}\right)\sqrt{n-1}\;\sim t_{n-1}.$$

Further, if n > 3, unconditionally,
$E\left({\hat{\beta }}_{1}\right)=\beta_{1}$ and Var$\left({\hat{\beta }}_{1}\right)=(\sigma^{2}/\sigma_{X}^{2})*{\left(n-3\right)}^{-1}$;In the conditional set-up,
$$E\left({\hat{\beta }}_{1}|X_{1},X_{2},\ldots ,X_{n}\right)=\beta_{1}$$
$$\mathrm{Var}\left({\hat{\beta }}_{1}|X_{1},X_{2},\ldots ,X_{n}\right)=\sigma^{2}/S_{XX};$$The random variable $U=\left({\hat{\beta }}_{1}-\beta_{1}\right)\sigma_{X}/\sigma$ has the probability density function:$$f\left(U\right)=\frac{1}{B\left(\frac{1}{2},\frac{n-1}{2}\right)}*\frac{1}{{\left(1+U^{2}\right)}^{\frac{n}{2}}},\;-\infty <U<\infty ;$$It follows that $U^{2}\sim W_{1}/W_{2}$, where $W_{1}\sim \chi_{1}^{2}$ and $W_{2}\sim \chi_{n-1}^{2}$, with $W_{1}$ and $W_{2}$ being independent;Exact distribution of $T^{2}$: note that $\left({\hat{\beta }}_{1}-\beta_{1}\right)/\sigma$ and ${\hat{\sigma }}_{X}^{2}$ are independent; $$T^{2}={\left({\hat{\beta }}_{1}-\beta_{1}\right)}^{2}{\hat{\sigma }}_{X}^{2}/{\hat{\sigma }}^{2}=\left\{{\left({\hat{\beta }}_{1}-\beta_{1}\right)}^{2}*\sigma_{X}^{2}/\sigma^{2}\right\}*(\sigma^{2}/{\hat{\sigma }}^{2})*{\hat{\sigma }}_{X}^{2}/\sigma_{X}^{2}\;\sim (W_{1}/W_{2})*\left\{\left(n-2\right)/W_{3}\right\}*W_{4}/\left(n-1\right),$$
where $W_{3}\sim \chi_{n-2}^{2}$ and $W_{4}\sim \chi_{n-1}^{2}$, and with $W_{1},W_{2},W_{3},$ and $W_{4}$ being independent.In short,
(3)$$T^{2}\;\sim \;\left(n-2\right)/\left(n-1\right)*(W_{1}W_{4})/(W_{2}W_{3})$$It follows that:$$E\left(T^{2}\right)=\left(n-2\right)/\{\left(n-3\right)\left(n-4\right)\;$$An alternative form of the distribution [22]: $$\left(n-1\right)/\left(n-2\right)*T^{2}\;\sim \;(W_{1}W_{4})/(W_{2}W_{3})\;\sim \;Beta_{II}\left\{\frac{1}{2},\frac{n-2}{2}\right\}*Beta_{II}\left\{\frac{n-1}{2},\frac{n-1}{2}\right\},$$
where $Beta_{II}$ signifies the beta distribution of the second kind. 

### 3.2. Critical Values 

We obtain the critical values of the test based on the test statistic $T={\hat{\beta }}_{1}{*\hat{\sigma }}_{X}/\hat{\sigma }$ for three levels of significance. We denote the critical value by $C_{n,\alpha }$. The critical value $C_{n,\alpha }$ satisfies the equation:$$\alpha =\mathrm{Pr}\left(\left|\frac{{\hat{\beta }}_{1}{\hat{\sigma }}_{X}}{\hat{\sigma }}\right|>C_{n,\alpha }|H_{0}:\beta_{1}=0\right)=\mathrm{Pr}\left(\frac{{\hat{\beta }}_{1}^{2}{\hat{\sigma }}_{X}^{2}}{{\hat{\sigma }}^{2}}>C_{n,\alpha }^{2}|H_{0}:\beta_{1}=0\right)$$

Under $H_{0}$,
$$T^{2}=\frac{{\hat{\beta }}_{1}^{2}{\hat{\sigma }}_{X}^{2}}{{\hat{\sigma }}^{2}}\sim \frac{\left(n-2\right)}{\left(n-1\right)}*\frac{W_{1}W_{4}}{W_{2}W_{3}},$$
where $W_{1}\sim \chi_{1}^{2}$, $W_{2}\sim \chi_{n-1}^{2}$, $W_{3}\sim \chi_{n-2}^{2}$ and $W_{4}\sim \chi_{n-1}^{2}$, with W_i_ values being independent. 

There are two options. One is using the pdf of $\left\{\left(n-2\right)/\left(n-1\right)\right\}$ ∗ T^2^. Following Jambunathan [22], one can write the pdf of the product U*V of the random variables U and V with U$\sim {\mathrm{Beta}}_{\mathrm{II}}\left(1/2\;,\left(n-2\right)/2\right)$, V$\sim {\;\mathrm{Beta}}_{\mathrm{II}}\left(\left(n-1\right)/2,\left(n-1\right)/2\right)$, and U and V being independent. The pdf is in the form of a double integral and its evaluation would require the use of a quadrature formula with the attendant errors of approximation. The second option is to determine the distribution of T^2^ by sampling extensively the components that make up T^2^ via Monte-Carlo. We have pursued the second option. The critical values are tabulated in Appendix A.

One can also obtain the critical value $C_{n,\alpha }$ via the asymptotic distribution of T. One benefit of our derivation of the exact distribution is that if n is large, and null hypothesis is true,
$$T\sim \mathrm{Normal}\;(0,\;\left(n-2\right)/\{\left(n-3\right)(n-4)\}),\;\mathrm{approximately}.$$

There are several ways to establish the asymptotic normality of T. The exact unconditional distribution of $\sqrt{n-1}$ $*\left({\hat{\beta }}_{1}-\beta_{1}\right){*\sigma }_{X}/\sigma$ is t_n−1_, which is asymptotically N (0, 1). Then we use the fact that ${\hat{\sigma }}_{X}$ is consistent for $\sigma_{X}$ and that $\hat{\sigma }$ is consistent for σ. Since we know the variance of T exactly, we use this variance in the description of the asymptotic distribution of T. We can now calculate the critical values, as well as those coming from the exact distribution, following the asymptotic distribution. 

In Appendix A, we report the average critical values C_n,α_ along with the critical values stemming from the asymptotic theory. A description of these asymptotic critical values is provided below. 

Critical values from the normal approximation:
LevelCritical Value FormulaVerbal description in Appendix A10%$1.645\times \sqrt{\left(n-2\right)/\left\{\left(n-3\right)\left(n-4\right)\right\}}$10% normal5%$1.96\times \sqrt{\left(n-2\right)/\left\{\left(n-3\right)\left(n-4\right)\right\}}$5% normal1%$2.576\times \sqrt{\left(n-2\right)/\left\{\left(n-3\right)\left(n-4\right)\right\}}$1% normal

Comments on Appendix A: The Normal Critical Value column is explained. 

Normal critical value 10% = critical value coming from the asymptotic distribution when α = 0.10.Normal critical value 5% = critical value coming from the asymptotic distribution when α = 0.05.Normal critical value 1% = critical value coming from the asymptotic distribution when α = 0.01.Critical value 10% = critical value coming from the exact distribution of T when α = 0.10.Critical value 5% = critical value coming from the exact distribution of T when α = 0.05.Critical value 1% = critical value coming from the exact distribution of T when α = 0.01.When α = 0.10, |Normal critical value 10%: Critical value 10%| ≤ 0.001 for n ≥ 50. One can enjoy the benefit of normal approximation when n ≥ 50.When α = 0.05, |Normal critical value 5%: Critical value 5%| ≤ 0.001 for n ≥ 89. One can enjoy the benefit of normal approximation when n ≥ 89.For α = 0.01, Table 1 is not informative when |Normal critical value 1%: Critical value 1%| ≤ 0.001.

### 3.3. Sample Size and Power

For a given level α, sample size n, and alternative value of β_1_ = A, power is given by
$$\mathrm{Power}\;(A)=\mathrm{Pr}(\left|{\hat{\beta }}_{1}{*\hat{\sigma }}_{X}/\hat{\sigma }\right|>C_{n,\alpha }\;|{\;\beta }_{1}=A).\;$$

Suppose 1 − β is the specified power. For the sample size, we set
$$1-\beta =\mathrm{Pr}(\left|{\hat{\beta }}_{1}{*\hat{\sigma }}_{X}/\hat{\sigma }\right|>C_{n,\alpha }\;|{\;\beta }_{1}=A).$$
and solve for n. We will need the distribution of ${\hat{\beta }}_{1}{*\hat{\sigma }}_{X}/\hat{\sigma }$, when $\beta_{1}=A$. Rewrite
$${\hat{\beta }}_{1}{*\hat{\sigma }}_{X}/\hat{\sigma }=({\hat{\beta }}_{1}-{\;\beta }_{1}){*\hat{\sigma }}_{X}/\hat{\sigma }+\beta_{1}{*\hat{\sigma }}_{X}/\hat{\sigma }.\;$$

The distribution of $({\hat{\beta }}_{1}-{\;\beta }_{1}){*\hat{\sigma }}_{X}/\hat{\sigma }$ is described in Section 3.1 and it is free of the parameters of the regression model. Consequently, the random variables $({\hat{\beta }}_{1}-{\;\beta }_{1}){*\hat{\sigma }}_{X}/\hat{\sigma }$ and $\beta_{1}{*\hat{\sigma }}_{X}/\hat{\sigma }$ are independently distributed. Since $\hat{\sigma }$ and ${\hat{\sigma }}_{X}$ are independently distributed,
$${\left(\beta_{1}{*\hat{\sigma }}_{X}/\hat{\sigma }\right)}^{2}\;\to^{d}\;\beta_{1}^{2}{*\sigma }_{X}^{2}/\left(n-1\right){*W}_{5}*\left(n-2\right)/\sigma^{2}*1/W_{6}\;={\left(\beta_{1}{*\sigma }_{X}/\sigma \right)}^{2}\left(n-2\right)/\left(n-1\right){*W}_{5}/W_{6},$$
with W_5_~$\chi_{n-1}^{2}$, W_6_~$\chi_{n-2}^{2}$, and W_5_ and W_6_ being independent. 

An important fact emerges from these deliberations in that the distribution of $({\hat{\beta }}_{1}-{\;\beta }_{1}){*\hat{\sigma }}_{X}/\hat{\sigma }$ + $\beta_{1}{*\hat{\sigma }}_{X}/\hat{\sigma }$ depends only on δ = $\beta_{1}{*\sigma }_{X}/\sigma$, which we declare as the effect size. 

In short, when β_1_ = A ≠ 0, the key steps are:
T = $({\hat{\beta }}_{1}-{\;\beta }_{1}){*\hat{\sigma }}_{X}/\hat{\sigma }$ + $\beta_{1}{*\hat{\sigma }}_{X}/\hat{\sigma }$with ${\left\{({\hat{\beta }}_{1}-{\;\beta }_{1}){*\hat{\sigma }}_{X}/\hat{\sigma }\right\}}^{2}$~$\left\{\left(n-2\right)/\left(n-1\right)\right\}*(W_{1}{*W}_{4})/(W_{2}{*W}_{3}),$$(\beta_{1}{*\hat{\sigma }}_{X}/\hat{\sigma }$)^2^~${\left({A*\sigma }_{X}/\sigma \right)}^{2}*\left(n-2\right)/\left(n-1\right){*W}_{5}/W_{6}$,$({\hat{\beta }}_{1}-{\;\beta }_{1}){*\hat{\sigma }}_{X}/\hat{\sigma }$ and $\beta_{1}{*\hat{\sigma }}_{X}/\hat{\sigma }$ are independent,
and the distribution of T depends only on n and effect size δ = $\left|A\right|{*\sigma }_{X}/\sigma$.

In spite of all these labors, the distribution of T is not amenable to direct and simple computation of power. 

We simulate the regression model for power computations. Simulations are greatly simplified when we exploit the key nature of the alternative distribution, namely, that it depends only on n and δ. Simulations are reported in the Appendix A. Sample sizes are tabulated in Table 1, Table 2 and Table 3.

**Table 1 entropy-25-00611-t001:** Sample Size for Given Effect Size, Power, Level of Significance 10%, Mean of Power in the Validation Step, and its Standard Deviation.

α	ES = β1 ∗ (σx/σ)	Power	n	Mean	Sd
0.1	0.1	80%	620	0.7993	0.013
		90%	870	0.9027	0.0095
		95%	1120	0.9546	0.0067
		99%	1690	0.993	0.0027
0.1	0.2	80%	161	0.8259	0.0123
		90%	219	0.90007	0.0096
		95%	274	0.949	0.0069
		99%	440	0.0039	0.0024
0.1	0.3	80%	73	0.8017	0.0124
		90%	100	0.9006	0.00995
		95%	124	0.9475	0.0071
		99%	195	0.931	0.0026
0.1	0.4	80%	43	0.8031	0.0126
		90%	60	0.9073	0.0093
		95%	72	0.9518	0.0073
		99%	105	0.9896	0.0031
0.1	0.5	80%	29	0.8045	0.0134
		90%	39	0.9003	0.0099
		95%	48	0.947	0.0068
		99%	69	0.989	0.0034
0.1	0.6	80%	21	0.8	0.0129
		90%	28	0.8961	0.0096
		95%	35	0.9476	0.0069
		99%	52	0.9911	0.003

Comments on Table 1, Table 2 and Table 3: The first column in each table entertains three types of effect sizes: small (0.1, 0.2); medium (0.3, 0.4); and large (0.5, 0.6) [15].The second column in each table lays out the powers entertained.The third column in each table spells out the requisite sample size.The fourth column is the fruit of our effort to validate the sample size. At the ascertained sample size, data are generated under the specifications, power calculated, and power averaged over thousand times.The fifth column records the standard deviation of the thousand powers calculated.We are satisfied that the sample sizes laid out are holding true. 

## 4. Discussion

A simple linear regression is a five-parameter model spelling out causality between two quantitative variables Y and X typified by:$$Y|X\sim N\left(\beta_{0}+\beta_{1}X,{\;\sigma }^{2}\right)\;X\sim N\left({µ}_{X},{\;\sigma }_{X}^{2}\;\right)$$
for some parameters $\beta_{0},{\;\beta }_{1},{\;\mu }_{X}$, $\sigma^{2}>0$, and $\sigma_{X}^{2}>0$. The goal is to sample (X, Y) for testing $H_{0}:\beta_{1}=0$ versus the alternative $H_{1}:\beta_{1}\neq 0$. For determining sample size, we need the level of significance α, power 1 − β, and the effect size $\delta ={A*\sigma }_{X}/\sigma$, where A is the given alternative value of $\beta_{1}$. The test statistic T used here is the one based on the least squares’ estimator ${\hat{\beta }}_{1}$ of ${\;\beta }_{1}$.

The regression model, as originally formulated, is a conditional model, i.e., $Y|X\sim N\left(\beta_{0}+\beta_{1}X,{\;\sigma }^{2}\right)$. In practice, in a planned experiment, the experimenter selects $x_{1},x_{2},.\;.\;.,x_{n}$ of X, and observes one or more Ys from the conditional distribution of Y|x_i_ for each i. Thus, the sample size n has already been chosen. The statistic $\left({\hat{\beta }}_{1}\sqrt{S_{\mathrm{XX}}}\right)/\left(\sqrt{\mathrm{RSS}/\left(n-2\right)}\right)$ is used for testing $H_{0}:\beta_{1}=0$ against the alternative $H_{1}:\beta_{1}\neq 0$. The conditional distribution of the test statistic given the data on X is Student’s t with n-2 degree of freedom under $H_{0}$, and the distribution is non-central Student’s t with n-2 degrees of freedom and non-centrality parameter $\left|A\right|*\sqrt{\mathrm{Sxx}}/\sigma$ under $H_{1}:\;\beta_{1}=A$. The alternative distribution can be used to calculate the power of the test at $\beta_{1}=A$, and nothing more. The entities n and $S_{\mathrm{XX}}$ are already in place, and σ has to be spelled out. The value of A is provided by the experimenter as the one of clinical significance. From the consulting experience of one of the authors, the experimenter usually comes up with value for σ from his/her pilot study.

In some statistical circles [6,7], the non = null distribution is used to calculate the sample size with the desired power, with $S_{\mathrm{XX}}$ remaining the same. This is controversial and discussed in [3,5,13].

We are dealing with unplanned experiments in which both X and Y are sampled together. Unplanned experiments are very common in clinical studies [4]. The effect size, in this context, is the multiple of the alternative value of $\beta_{1}$ by the ratio of the two standard deviations of the model.

The current practice demands α, 1 − β, A, σ and $S_{\mathrm{XX}}$, which we do not have. Specification of $S_{\mathrm{XX}}$ is avoided by determining the unconditional distribution of
$$T={\hat{\beta }}_{1}*{\hat{\sigma }}_{X}/\hat{\sigma }.$$

Exploiting the unconditional distribution of T, we calculated the critical values and required sample size. The unconditional distribution under the alternative depends on the effect size δ = $\beta_{1}{*\sigma }_{X}/\sigma$, as well as n and α. As a contrast, popular software such as PASS [6] and nQuery [7] use the conditional distribution of the test statistic T^*^ given the data on X, for calculating sample size. 

An additional feature of our paper is that we provide a comprehensive table of critical values and sample sizes, unlike commercial software. 

The main result that the non-null distribution of the test statistic T depends only on the effect size δ has an echo in other inference problems. For example, when testing $\mu_{1}=\mu_{2}$ under the normality and common variance $\sigma^{2}$ assumptions, the non-null distribution of the two-sample t-statistic depends only on the effect size $\lambda =(\mu_{1}-\mu_{2)}/\sigma$. This result, in spirit, is like ours. We have archived our findings for comments and insights [23]. 

We trust that the tables provided will help researchers to calculate sample size in the context of simple linear regression in unplanned experiments avoiding the controversies that have been problematic till now. We will continue to study how sample sizes are contrasted between one test based on the slope parameter of the model and one based on the correlation coefficient. 

## Figures and Tables

**Table 2 entropy-25-00611-t002:** Sample Size for Given Effect Size, Power, Level of Significance 5%, Mean of Power in the Validation Step, and its Standard Deviation.

α	ES = β1 ∗ (σx/σ)	Power	n	Mean	Sd
0.05	0.1	80%	790	0.8006	0.0129
		90%	1080	0.9054	0.0088
		95%	1350	0.9557	0.0067
		99%	1850	0.9898	0.0032
0.05	0.2	80%	199	0.797	0.0133
		90%	272	0.9039	0.0094
		95%	330	0.9497	0.0069
		99%	450	0.9891	0.0033
0.05	0.3	80%	91	0.7978	0.0124
		90%	123	0.9028	0.0094
		95%	150	0.9505	0.0067
		99%	220	0.992	0.0028
0.05	0.4	80%	53	0.773	0.0128
		90%	70	0.8966	0.0096
		95%	87	0.9494	0.0071
		99%	121	0.9891	0.0034
0.05	0.5	80%	36	0.8051	0.0124
		90%	48	0.9095	0.0091
		95%	58	0.95	0.0068
		99%	79	0.9888	0.0033
0.05	0.6	80%	26	0.8005	0.0124
		90%	34	0.8985	0.0094
		95%	43	0.9547	0.0066
		99%	59	0.9901	0.0031

**Table 3 entropy-25-00611-t003:** Sample Size for Given Effect Size, Power, Level of Significance 1%, Mean of Power in the Validation Step, and its Standard Deviation.

α	ES = β_1_ ∗ (σx/σ)	Power	n	Mean	Sd
0.01	0.1	80%	1180	0.8026	0.0124
		90%	1500	0.9015	0.0095
		95%	1760	0.946	0.0072
		99%	2440	0.9906	0.0031
0.01	0.2	80%	301	0.8045	0.0121
		90%	388	0.9046	0.0093
		95%	458	0.9529	0.0065
		99%	620	0.991	0.0031
0.01	0.3	80%	136	0.8044	0.0129
		90%	172	0.9012	0.0089
		95%	199	0.9432	0.0071
		99%	265	0.9872	0.0034
0.01	0.4	80%	78	0.8017	0.0124
		90%	95	0.8856	0.0099
		95%	118	0.949	0.007
		99%	158	0.9892	0.0033
0.01	0.5	80%	51	0.8042	0.0126
		90%	64	0.9011	0.0099
		95%	77	0.9464	0.007
		99%	104	0.9891	0.0032
0.01	0.6	80%	37	0.7975	0.0125
		90%	48	0.906	0.0089
		95%	56	0.9485	0.007
		99%	73	0.9874	0.0035

## Data Availability

Not applicable.

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
