# Peer review of "Sample Size Calculations in Simple Linear Regression: A New Approach"

_entropy, 2023, doi:10.3390/e25040611_

Round 1

Reviewer 1 Report

Dear [Author],

Thank you for the opportunity to review your manuscript titled " Sample Size Calculations in Simple Linear Regression: A New Approach" . After carefully reading and analyzing the manuscript, I regret to inform you that I cannot recommend it for publication in its current form.

My main concern with the paper is that it lacks novelty and significance. Although the topic of simple regression is important, your paper does not present any new additions to the literature or offer any innovative approaches. Rather, the work appears to be routine and limited to a specific case.

Furthermore, I found some weaknesses in the methodology used in the study. For example,there is no application in the paper, and the statistical analysis performed lacks sufficient detail and justification. Additionally, the paper lacks clarity in some areas, which makes it difficult to understand and evaluate the methods and results.

Based on these concerns, I suggest that you revise the paper thoroughly to address the aforementioned issues. You may consider expanding the scope of the research to include more advanced methods or novel approaches. Additionally, I recommend that you address the limitations in the methodology and provide a more detailed and clear explanation of your methods and results.

Overall, I believe that your research has the potential to contribute to the field, but it needs significant revisions before it can be considered for publication. I encourage you to take my feedback into account and submit a revised version of the manuscript after making appropriate changes.

Thank you again for the opportunity to review your work, and I wish you the best in your future research.

Author Response

Dear reviewer,

Here are our responses to your points:

Point 1: My main concern with the paper is that it lacks novelty and significance. Although the topic of simple regression is important, your paper does not present any new additions to the literature or offer any innovative approaches. Rather, the work appears to be routine and limited to a specific case.

Response to Point 1:

We have added a new paragraph in discussion explaining the irrationality of the existing methods.  We have enunciated our motivation for this work more clearly.

Point 2: Furthermore, I found some weaknesses in the methodology used in the study. For example, there is no application in the paper, and the statistical analysis performed lacks sufficient detail and justification. Additionally, the paper lacks clarity in some areas, which makes it difficult to understand and evaluate the methods and results.

Response to Point 2:

Simple linear regression is prominent in clinical diagnostics. An example is discussed in the introduction. The current methodology used in determining sample size is faulty. Its irrationality is discussed in Discussion section last paragraph. The contrast between planned and unplanned experiments is explained in Discussion.

Point 3: Based on these concerns, I suggest that you revise the paper thoroughly to address the aforementioned issues. You may consider expanding the scope of the research to include more advanced methods or novel approaches. Additionally, I recommend that you address the limitations in the methodology and provide a more detailed and clear explanation of your methods and results.

Response to Point 3:

The paper is revised substantially bringing more clarity.  

Point 4: Overall, I believe that your research has the potential to contribute to the field, but it needs significant revisions before it can be considered for publication. I encourage you to take my feedback into account and submit a revised version of the manuscript after making appropriate changes.

Response to Point 4:

Please review the revised version. We will be glad to respond to any comments and suggestions you make.

Thank you again for reviewing our work,

Tianyuan Guan

Reviewer 2 Report

In this paper, the authors considered a five-parameter model of simple linear regression and discussed the distribution of least square estimator hat{beta}_1. Under the sample of bivariate normal distribution, the distribution of T^2= [hat{beta}_1*hat{sigma}_X/hat{sigma}]^2 was obtained. In order to show the critical values, some simulations were taken on at different levels. The sample size and power were also discussed. The works are important and interesting in the regression analysis. The limitation may be taken on the normal sample. So it is interesting for researchers to generate the results to non-normal sample and study the limiting distribution of least square estimation. I recommend it to publish in Entropy. Before publication, the authors should pay attention on the following problems.

(1)     In the introduction and discussion, some parameters were in bold. Usually, parameter in bold means the vector parameter, but in this paper, the parameters are one-dimensional parameters.

(2)     On page 3, line 75, equation number (1) should be right-aligned; line 97 has the same problem; line 106, [3, 15, 16, 14, 17, 18, 19, 20, 21] should be [3,14-21].

(3)     On page 4, line 121, “where” should be in top frame.

(4)     On page 5, line 161, U^2---d-->W_1/W2 may be U^2 ~ W_1/W2; line 165 has the same problem.

(5)     In reference, some journal were full names, however, some were abbreviations.

Author Response

Dear Reviewer,

Thanks for your perceptive comments and suggestions. Here is our point by point response.

  1. The parameters and random variables are not in Italics and bold anymore.
  2. The reference is stream-lined as suggested.
  3. The equations are right aligned as suggested.
  4. We used the "~" symbol to indicate "distributed as".
  5. Full names of Journals are provided uniformly.

Additional point: A detailed paragraph is included in the introduction to provide more clarity to the problem tackled in the paper.

Reviewer 3 Report

According to description of the article, a new method for sample size calculation of simple linear regression is proposed. But the quality of the article is not enough to publish.

In particular:

1. The motivation of the article is not clear, and the irrationality of the existing methods needs to be explained.

2. What are the main innovations of the new method compared with the existing method? The idea of solving the problem needs a simple and clear summary.

3. The writing format of the article is chaotic, especially the italic, bold and labeling of formula symbols.

4. The conclusion of the article is unclear and needs to be summarized. The conclusion needs strong results as support.

5. The references of the article are basically 10 years ago. Are there no references in recent years?

Author Response

Dear reviewer,

Thank you so much for your time and comments, here are our response to your points:

Point 1: The motivation of the article is not clear, and the irrationality of the existing methods needs to be explained.

response to point 1: We have added a new paragraph in discussion explaining the irrationality of the existing methods.  We have enunciated motivation more clearly.

Point 2: What are the main innovations of the new method compared with the existing method? The idea of solving the problem needs a simple and clear summary.

Response to point 2: 

The same paragraph in discussion addresses the criticism you raised.

Point 3: The writing format of the article is chaotic, especially the italic, bold and labeling of formula symbols.

Response to point 3:

The parameters and random variables are not in Italics and bold anymore. Full names of Journals are provided uniformly now. The equations are right aligned as suggested.

Point 4: The conclusion of the article is unclear and needs to be summarized. The conclusion needs strong results as support.

Response to point 4:

We edited and summarized the conclusions clearly in the last paragraph in the discussion. Please review.

Point 5: The references of the article are basically 10 years ago. Are there no references in recent years?

Response to point 5:

The irrationality of the existing methods persisted for a long time [Ryan, 2013]. We have aired this problem clearly in this article. We are not rectifying the method. We are working in a similar scenario with a clear cut and rational solution with innovative derivations of relevant distributions.  

Please review the revised manuscript,

Tianyuan

Reviewer 4 Report

See attached report.

Author Response

Dear reviewer,

Thank you for your good comments and questions! The following are my responses for your specific comments:

point 1. line 105-106 Is it really a choice between causality and association? Causality
only arises when the xi values are chosen and assigned to experimental units.

Response to point1:

It is a good point. Whether the experiment is planned or not, the estimate of the slope is interpreted in the same way. The causality idea is implied in the interpretation. Association is hard to interpret.

Point 2: 191-193 In other words you used Monte Carlo, correct? Please be more explicit about this. Also, I think you use Monte Carlo for the power calculations.

Response to point 2: 

Yes! We did Monte Carlo simulations. It is much easier to do simulations from the explicit description of the distribution of T2. We have acknowledged it in this paragraph.

Point 3: You should say something about why effect size is defined the way it is for power calculations. In other words, why is this the right way to define effect
size in an application?

Response to point 3: 

When we determined the alternative distribution, we observed that the distribution depends only on (ß1x)/σ, and sample size. It is natural to call (ß1x)/σ as effect size. This contrasts well with several examples. For example, the non-null distribution of two sample t-statistic t depends on the effect size (μ- μ2)/σ. We had it explained in Discussion section, paragraph 6.

Point 4. You mention an alternative approach to a power analysis involving the correlation coefficient but there really is no comparison between the two. You need to say more about this. Why is your approach preferred? Right now it isn't
very convincing.

Response to point 4:

The hypothesis H0: ß1=0 and H0: ρ=0 under bivariate normality are equivalent, but the test statistics are different, it is easy to determine sample size under correlation context [14]. However, the sample size cannot be offered for testing the hypothesis on the slope. The power is less. In other words, test hopping is not permissible.  

Point 5. What if the xi are not normally distributed? Can you do something similar in that case?

Response to point 5:

Normality assumptions solves many intricate distribution problems. In the non-normal case, one has to rely on the asymptotic theory. This will be a whole lot of new work.

Tian

Round 2

Reviewer 1 Report

I accept in current form

Author Response

Dear reviewer,

Thanks for accepting the revised version. 

MB

Reviewer 3 Report

ok with the revisions. 

Author Response

(The authors gave the same response as above.)

Reviewer 4 Report

l. 81 (spelling) This requires a knowledge of the distribution of X. Let $?_?^ ?$ be the variance of X.

1. 121 "test hopping is not permissible" What is "test hopping"? I think I know what you are getting at, namely, they are just different tests with different power functions. So why not say it explicitly.

l. 206 I think you want to say "The second option is to determine the distribution of T2 by sampling extensively the components that make up T2 via Monte-Carlo. 

Author Response

Dear reviwer,

Point1: l. 81 (spelling) This requires a knowledge of the distribution of X. Let $?_?^ ?$ be the variance of X.

Response to point 1: The spelling is corrected. 

Point 2: 1. 121 "test hopping is not permissible" What is "test hopping"? I think I know what you are getting at, namely, they are just different tests with different power functions. So why not say it explicitly.

Response to Point 2:

Test hopping is explained in the manuscript following your wording.

Point 3: l. 206 I think you want to say "The second option is to determine the distribution of T2 by sampling extensively the components that make up T2 via Monte-Carlo. 

Response to Point 3: The sentence is reframed following your wording.

Thanks,

MB